# Safety and Activity of the Combination of Ceritinib and Dasatinib in Osteosarcoma

**DOI:** 10.3390/cancers12040793

**Published:** 2020-03-26

**Authors:** Olaf Beck, Claudia Paret, Alexandra Russo, Jürgen Burhenne, Margaux Fresnais, Kevin Steimel, Larissa Seidmann, Daniel-Christoph Wagner, Nadine Vewinger, Nadine Lehmann, Maximilian Sprang, Nora Backes, Lea Roth, Marie Astrid Neu, Arthur Wingerter, Nicole Henninger, Khalifa El Malki, Henrike Otto, Francesca Alt, Alexander Desuki, Thomas Kindler, Joerg Faber

**Affiliations:** 1Department of Pediatric Hematology/Oncology, Center for Pediatric and Adolescent Medicine, University Medical Center of the Johannes Gutenberg-University Mainz, 55131 Mainz, Germany; Olaf.Beck@unimedizin-mainz.de (O.B.); Claudia.Paret@unimedizin-mainz.de (C.P.); Alexandra.Russo@unimedizin-mainz.de (A.R.); Nadine.Vewinger@unimedizin-mainz.de (N.V.); Nadine.Lehmann@unimedizin-mainz.de (N.L.); Maximilian.Sprang@unimedizin-mainz.de (M.S.); Nora.Backes@unimedizin-mainz.de (N.B.); Lea.Roth@unimedizin-mainz.de (L.R.); Marie.Neu@unimedizin-mainz.de (M.A.N.); Arthur.Wingerter@unimedizin-mainz.de (A.W.); Nicole.Henninger@unimedizin-mainz.de (N.H.); Khalifa.ElMalki@unimedizin-mainz.de (K.E.M.); Henrike.Otto@unimedizin-mainz.de (H.O.); Francesca.Alt@unimedizin-mainz.de (F.A.); 2University Cancer Center (UCT), University Medical Center of the Johannes Gutenberg-University Mainz, 55131 Mainz, Germany; Alexander.Desuki@unimedizin-mainz.de (A.D.); Thomas.Kindler@unimedizin-mainz.de (T.K.); 3German Cancer Consortium (DKTK), site Frankfurt/Mainz, Germany, German Cancer Research Center (DKFZ), 69120 Heidelberg, Germany; 4Department of Clinical Pharmacology and Pharmacoepidemiology, Heidelberg University Hospital, 69120 Heidelberg, Germany; Juergen.Burhenne@med.uni-heidelberg.de (J.B.); Margaux.Fresnais@med.uni-heidelberg.de (M.F.); Kevin.Steimel@med.uni-heidelberg.de (K.S.); 5German Cancer Consortium (DKTK)-German Cancer Research Center (DKFZ), 69120 Heidelberg, Germany; 6Institute of Pathology, University Medical Center of the Johannes Gutenberg-University Mainz, 55131 Mainz, Germany; Larissa.Seidmann@unimedizin-mainz.de (L.S.); Daniel-Christoph.Wagner@unimedizin-mainz.de (D.-C.W.); 7Department of Hematology, Medical Oncology, and Pneumology, University Medical Center of the Johannes Gutenberg-University Mainz, 55131 Mainz, Germany

**Keywords:** osteosarcoma, ceritinib, dasatinib

## Abstract

Osteosarcoma (OS) is the second most common cause of cancer-related death in pediatric patients. The insulin-like growth factor (IGF) pathway plays a relevant role in the biology of OS but no IGF targeted therapies have been successful as monotherapy so far. Here, we tested the effect of three IGF specific inhibitors and tested ceritinib as an off-target inhibitor, alone or in combination with dasatinib, on the proliferation of seven primary OS cells. Picropodophyllin, particularly in combination with dasatinib and the combination ceritinib/dasatinib were effective in abrogating the proliferation. The ceritinib/dasatinib combination was applied to the primary cells of a 16-year-old girl with a long history of lung metastases, and was more effective than cabozantinib and olaparib. Therefore, the combination was used to treat the patient. The treatment was well tolerated, with toxicity limited to skin rush and diarrhea. A histopathological evaluation of the tumor after three months of therapy indicated regions of high necrosis and extensive infiltration of macrophages. The extension of the necrosis was proportional to the concentration of dasatinib and ceritinib in the area, as analysed by an ultra performance liquid chromatography–tandem mass spectrometer (UPLC-MS/MS). After the cessation of the therapy, radiological analysis indicated a massive growth of the patient’s liver metastases. In conclusion, these data indicate that the combination of ceritinib/dasatinib is safe and may be used to develop new therapy protocols.

## 1. Introduction

Osteosarcoma (OS) is the second-leading cause of cancer-related death in pediatric patients and the most common primary bone tumor. The current management strategy for newly diagnosed osteosarcoma includes neoadjuvant chemotherapy followed by the surgical removal of the primary tumor along with all clinically evident metastatic disease, plus the addition of adjuvant chemotherapy after surgery [1,2,3]. Despite these multimodal therapeutic treatments, numerous patients develop recurrent or metastatic disease and new molecular strategies are urgently needed [4]. The development of targeted therapies has been not successful so far. This is mainly due to the rarity and the high genetic heterogeneity of these tumors, which makes it difficult to recruit patient cohorts that are large enough to compensate for high genetic variability [5]. Even if the complex genetic heterogeneity of OS renders targeted agents unlikely to succeed as monotherapy, very few combinations have been tested so far and no Phase-II trials have transposed to successful first-line Phase-III trials in the last three decades [6].

The insulin-like growth factor (IGF) pathway plays a relevant role in the biology of OS. Overexpression [7] and mutations [8] in members of the IGF pathway in OS have been described but neither small molecule inhibitors nor monoclonal antibodies have been approved by the US Food and Drug Administation (FDA) so far. However, we have recently shown that the IGF pathway can be targeted in vitro and in vivo using ceritinib [9,10]. Ceritinib is approved for the treatment of patients with Anaplastic lymphoma kinase (ALK)-positive metastatic non-small cell lung cancer (NSCLC) but can also inhibit the insulin receptor (INSR), the insulin-like growth factor 1 receptor (IGF1R) and ROS proto-oncogene 1 (ROS1) [11].

The currently available IGF1R-targeted therapies have shown marginal efficacy in advanced clinical trials for OS and other tumor entities, presumably due to the activation of bypass signaling as a resistance mechanism. Previous reports have suggested that proto-oncogene tyrosine-protein kinase Src (Src) can mediate resistance to IGF1R inhibition in rabdomyosarcoma [12] and targeting Src with dasatinib sensitizes NSCLC cells to IGF1R tyrosine kinase inhibitors (TKIs) [13]. Src is phosphorylated in OS cells [14] and inhibition of Src activity by dasatinib induced apoptosis in OS cell lines and inhibited cell motility and invasiveness in vitro [15]. A synergistic effect of dasatinib (as Src inhibitor) and ceritinib (as IGF1R inhibitor) has been described in rabdomyosarcoma cell lines [12] and one phase I/II trial is recruiting rabdomyosarcoma patients to investigate the combined effects of the IGF1R antibody ganitumab and dasatinib (NCT03041701). However, the combination of ceritinib and dasatinib has not been tested in clinical studies so far.

Functional testing of viable patient tumor cells exposed to potential therapies is emerging as a tool to define personalized therapy protocols [16]. Here, we performed a pharmacological screening in primary OS cell cultures identifying the combination of ceritinib and dasatinib as promising inhibitor of OS cell growth. This combination was applied to a pediatric patient with OS and a long history of lung metastases, leading to a histologic response with an acceptable toxicity profile.

## 2. Results

### 2.1. Drug Screening Identifies Ceritinib and Dasatinib as Potent Inhibitors of OS Proliferation In Vitro

To identify effective IGF inhibitors, we used primary tumor cells at low passage isolated from six patients (Table 1). OS primary cells were obtained from the vital area of the tumor after surgery and underwent morphological and immunocytochemical characterisation (Appendix A). The cell line HOS was also included in the analysis. We tested three IGF1R kinase inhibitors, namely, linsitinib [17], picropodophyllin (PPP) [18], and PQ401 [19] and ceritinib as off-target inhibitor of IGF1R and INSR [11], alone or in combination with dasatinib. As shown in Figure 1a, dasatinib, linsitinib, and PQ401 alone were effective in inhibiting the proliferation in some samples only. Accordingly, linsitinib has been shown to be particularly effective in some bone tumors derived cell lines in a pharmacologic high throughput screening (Appendix A). PPP alone completely abrogated the cell proliferation (ratio ≤ 1) in 50% of the samples. The combination of PPP with dasatinib and of ceritinib with dasatinib completely abrogated the cell proliferation in 75% and in 50% of the samples respectively. The efficacy of linsitinib and PQ401 was only slightly improved by the combination with dasatinib. Notably, we selected clinical relevant concentrations for our drug screening. A concentration of ceritinib, PPP, and linsitinib > 1 µM can be achieved in the plasma of patients [20,21,22]. In children with relapsed or refractory leukaemia, the maximum plasma concentration of dasatinib is >100 nM [23]. No data on the plasma concentration of PQ401 in patients is available to our knowledge and a concentration > 1 µM may be necessary to inhibit the IGF Pathway. PPP has not been released by the FDA or the European Medicines Agency (EMA) so far, and no current clinical studies with PPP are enrolling pediatric OS patients. For these reasons, we prioritized the ceritinib/dasatinib combination for further analyses. The ceritinib/dasatinib combination was tested on the OS primary tumor cells of a female 16-year-old patient with a long history of lung metastases (Sample 386). After several tumor resections and multimodal chemotherapies, the standardized conventional treatment options were used up for this patient. There were no established therapeutic concepts that could cure the disease in recurrence. However, the patient was still in an excellent clinical condition and had a firm will to make use of another therapy trial in a novel approach. Because BRCAness has been suggested as potential target in OS [24] and cabozantinib has shown promising results in OS [25], we tested also olaparib alone and in combination with irinotecan and cabozantinib on the primary cells of the patient. As shown in Figure 1b, the combination of ceritinib and dasatinib was the most effective. Moreover, IGF2 stimulation induced INSR phosphorylation and activation of AKT serine/threonine kinase 1 (AKT), while Src was already phosphorylated (Figure 1c). ALK, the main target of ceritinib, was not mutated, as assessed by the Sanger sequencing of the coding region. In conclusion, these data indicate that PPP alone is effective as a single IGF inhibitor and its activity can be improved by adding dasatinib. Dasatinib also increases the efficacy of ceritinib, which is not effective as a single agent.

### 2.2. Dosage of Dasatinib and Ceritinib in a Therapy Protocol

Based on our results, we planned a therapy within the framework of an individual healing attempt with ceritinib and dasatinib. The application of ceritinib and dasatinib was discussed and approved in the local molecular tumor board and was performed under strictly controlled conditions over a period of three months. The therapy monitoring and the response were documented weekly by blood analysis, cardiological and physical examination and after six to eight weeks by computed tomography (CT) and an ultrasound scan. Drug–drug interactions were evaluated using drug interaction software AiDKlinik^®^ version 2.0 1.4.19; (Dosing GmbH, Heidelberg, Germany). Both dasatinib and ceritinib are substrates and inhibitors of CYP 3A4. Studies have shown that the area under the curve (AUC) of both active substances can be increased by a factor of two to five in combination. Therefore, the resulting interaction and side effect risk could not be quantified in advance, particularly due to the fact that both agents may cause QT interval time prolongation. Taking into account the possible risk of side effects, a reduced dose of therapy was initially scheduled. Depending on the clinical response, an increase in dose was planned and the levels of the active substances in the blood were determined. With regard to the use of ceritinib in children, an acceptable tolerability of the drug at a dose of 510 mg/m² was observed in a study with 22 patients with an ALK-positive tumour [26]. Ceritinib was taken with a meal. In combination with a CYP 3A4 inhibitor, a dose reduction of 30% is recommended according to the manufacturer’s information. The dosage of dasatinib for children and adolescents with Philadelphia chromosome-positive chronic myeloid leukemia (Ph+ CML) or Philadelphia chromosome-positive acute lymphoblastic leukemia (Ph+ ALL) and a weight of at least 45 kg is recommended to be 100 mg once daily based on better dasatinib tolerability with once-daily dosing [27]. If strong CYP3A4 inhibitors are used at the same time, a dose reduction of 20 mg should be considered (according to local decision). Initially, the patient received a daily dose of 100 mg dasatinib followed by ceritinib 150 mg on day 10. According to the tablet size of ceritinib, a weekly dose increase of 150 mg per day was attempted.

Already with a dose of 100 mg dasatinib in combination with 300 mg ceritinib a local limited skin rash of several weeks as well as single episodes with diarrhoea occurred. However, these could be well treated locally or symptomatically, resulting in a short-term dose reduction (two weeks in total) of ceritinib from 300 mg to 150 mg only twice. Therefore, before a further increase of ceritinib, dasatinib was reduced to 80 mg/day in week five of therapy. There were no further side effects, so ceritinib could be increased to 450 mg per day during the course of treatment, while continuing to take 80 mg dasatinib per day. Fortunately, there was no interruption of therapy necessary at any time. Even if side effects led to individual dose adaptions, the patient could be treated on an outpatient basis during the therapy of almost three months. Inpatient admission to the clinic was not necessary.

Plasma levels were regularly measured during the therapy by UPLC-MS/MS. Since dasatinib was taken in the evening and the blood sample was often taken in the morning (after 10–14 h), a “medium” level of 20–32 ng/mL (41–65 nM) could be detected. Directly before taking the medication, a trough level of 6 ng/mL (12 nM) was measured. Despite the reduced dose of 80 mg, a peak level of 64 ng/mL (131 nM) was determined within four hours after taking the medication (Figure 2). There were no indications of an accumulation of dasatinib. Taking ceritinib, a plasma level of 394 ng/mL (706 nM) was measured after 10 h. Since ceritinib was usually administered in the morning after blood collection, a trough level could be regularly determined. Even 24 h after the last intake, a level of approximately 300–350 ng/mL was measured (Figure 2).

Under therapy, there was an increase in lactate dehydrogenase (LDH) to a maximum of 412 U/L and slight changes in the blood count with leukopenia of 3.42/nL and a drop in platelets to 79/nL. We had no sign of neutropenia or change in blood glucose level. After the reduction of dasatinib to 80 mg/day, the blood count normalized rapidly. No further abnormalities were found in the blood tests. Weekly electrocardiogram and regular echocardiography were normal.

In conclusion, the combination of dasatinib and ceritinib can be applied in a therapy protocol with acceptable toxicity.

### 2.3. Activity of the Certinib/Dasatinib Combination

After 56 days from the beginning of the therapy, a CT scan showed a progress in the lung and a new lesion in the liver. After 91 days from the beginning of the therapy, the tumor mass in the lung was eliminated by surgery. Pathological analysis indicated several areas of necrosis (Figure 3 upper row) and massive infiltration of macrophages (Figure 3 lower row and Table 2). The concentrations of ceritinib and dasatinib were measured in four regions of the tumor by UPLC-MS/MS. As shown in Table 2, a strict correlation was observed between the amount of the drugs in the tissue and the amount of necrosis. In the area with the lowest amount of vital tumor cells a concentration of 52.9 ng/g (108 nM) dasatinib and 1703 ng/g (3052 nM) ceritinib was detected. In the area with the highest amount of vital tumor cells a concentration of 687 ng/g ceritinib (1231 nM) was measured, while no dasatinib was detectable. Primary cells isolated from the vital area (sample no 403) were still responsive to the combination ceritinib/dasatinib but not to the single agents (Appendix A).

After surgery for pulmonary metastases, the patient (initially ventilated) was in a very weakened general condition and the oral combination therapy could not be taken. The liver metastases increased rapidly (Figure 4). Two months after surgery the patient died.

## 3. Discussion

This is the first report on the clinical application of the combination of ceritinib and dasatinib. Even if Src and the IGF pathway have shown initially positive results as therapeutic targets, neither of them have been recognized as a clinically effective target so far. Dasatinib as a monotherapy has shown limited clinical efficacy in a Phase-II study in sarcoma [30]. The inhibition of the IGF1R signaling alone using the monoclonal antibody R1507 has shown only partial response in a Phase-II trial [31]. These clinical results are in accordance with our preclinical results showing only limited action of the single agents.

Concerning the toxicity, the combination was well tolerated only with local limited skin rash and single episodes with diarrhea. In particular, no cardiac toxicity was observed. Concerning the efficacy of the treatment, we did not observe a reduction of the tumor mass by CT after two months, however, tumor volume shrinkage according to the response evaluation criteria in solid tumors (RECIST) criteria may not correctly reflect drug efficacy in osteosarcoma due to the replacement of the tumor mass by necrotic tissue and no apparent volume reduction. In bone lesions, chemotherapy, proven to improve overall survival, does not result in radiographic responses as measured by RECIST [32] and RECIST-based stable/progressive disease is not associated with tumor necrosis, a well-established prognostic factor in osteosarcoma [2]. Moreover, tumour shrinkage with targeted therapies may be delayed or absent, even if the drug is biologically efficient and some tyrosine-kinase inhibitors have been shown to provide patient benefit even after RECIST-defined progression, such as EGFR inhibitors in non-small cell lung cancer [33]. Accordingly, after surgery, the tumor area isolated from different localizations were highly necrotic, with a strict correlation between the extension of necrosis and the concentration of the dasatinib and ceritinib concentration. Necrosis has been previously described as associated with the response to IGF1R inhibitors in relapsed malignant astrocytoma and squamous non-small cell lung carcinoma [34,35] and after treatment with ceritinib as an IGF1R inhibitor in a malignant neuroepithelial brain tumor [9]. Tumor volume might even temporarily increase due to immune cell infiltration [36]. Indeed, our analysis showed a massive infiltration of macrophages in accordance with previous data identifying macrophages as the most abundant cell population in immune infiltrates of OS [37]. During the ceritinib/dasatinib therapy, new lesions in the liver were observed. Liver metastases are rare in OS and it has been previously discussed that their detection may be due to the improvement of systemic treatment, because the prolonged survival time allows OS cells to manifest in uncommon sites [38]. Importantly, the liver metastases were slow-growing under the ceritinib/dasatinib therapy, but started to grow impressively after the cessation of the therapy. These data, i.e., the extensive necrosis correlated to the drug penetration and the massive growth of the metastases after the stop of the therapy, suggest a role of the combination on the biology of the disease. Thus, in future, other tools for efficacy measurement and other time schedules to monitor the volume mass should be taken into consideration to assess the response to the therapy. It should be mentioned that the dose limitation was not achieved in this patient and the usage of high concentration or the simultaneous application of the two drugs could even improve the clinical benefit.

It remains to be clarified why the drugs’ penetrations were different in different areas of the tumor and if this represents a mechanism of resistance to the therapy. Primary tumor cells isolated from the vital area of the tumor after the combination therapy were still responsive in vitro to the treatment. Several factors may have influenced the uptake of the drugs in vivo, such as the expression of efflux proteins, the massive presence of immune cells or a reduced blood supply.

Our results also indicate the relevance of PPP alone and particularly in combination with dasatinib in treatment of OS. PPP has been shown to suppress the proliferation of osteosarcoma cell lines by inhibiting the IGF1R pathway and to reverse their drug-resistant phenotype [39]. PPP has shown an acceptable safety profile and demonstrated promising efficacy in heavily pretreated patients with advanced solid tumors, especially in patients with NSCLC [35]. However, PPP has not been approved so far.

The main reason for testing ceritinib and PPP in this study was to target the IGF signaling. However, the mechanisms leading to the IGF activation in the primary tumour cells we used are still unidentified. Indeed, the IGF Pathway activation may be dependent not only on the amplification/overexpression/ mutation of IGF1R and INSR receptors or overexpression of IGF1 and IGF2, but also on the downregulation of IGF-binding proteins (IGFBP) [40]. Moreover, multiple membrane-associated receptors can phosphorylate IGF1R via Src activation [13]. Finally, we can not exclude that the mechanism of action of ceritinib and PPP in osteosarcoma is independent of IGF. Ceritinib can inhibit also ROS1. In the case of PPP, additional effects unrelated to the effect on IGF1R signaling have been suggested [41]. Our results indicate that INSR phosphorylation can be induced by IGF2 in the primary cells of the patient we treated in this study, and that ALK was not mutated, however, further experiments are necessary to clarify the activation of the IGF pathway and the response to ceritinib/dasatinib in OS.

In this study, we applied the combination of ceritinib with dasatinib to a patient with a long history of lung metastasis. Pulmonary metastasis is a main cause of death for OS patients, and to prevent the spread of OS cells to the lung would improve OS survival rate. Dasatinib alone has been shown to reduce the migration of OS cells by blocking focal adhesion kinase (FAK) and Crk-associated substrate (p130CAS) signaling downstream of Src in sarcoma cell lines [15]. The question of whether the combination of ceritinib and dasatinib even improves this effect remains to be elucidated.

The relevance of the combination of ceritinib and dasatinib has been suggested also for the main target of ceritinb, namely ALK. Indeed, targeting Src signaling may be an effective approach to the treatment of ALK-NSCLC with acquired resistance to ALK inhibitors [42].

## 4. Materials and Methods

### 4.1. Tumor Cells

Isolation of primary tumor cells was performed by a pathologist from fresh tumor material using the cryosections as control to identify the regions containing vital OS cells. The cells were isolated by mechanical tissue dissocation with GentleMACS dissociator (Miltenyi Biotec, Bergisch-Gladbach, Germany) followed by enzymatical tissue dissocation with 400 µL Liberase^TM^ Research Grade (Roche, Basel, Switzerland), 3.6 mL HBSS medium (Thermo Fisher Scientific, Waltham, MA, USA) and 8 µL DNase I (Sigma-Aldrich, St. Louis, MO, USA). Isolated cells were cultured in Dulbecco’s Modified Eagle’s Medium (DMEM) with 10% human serum, 1% L-glutamine, and 1% Penicillin-Streptomycin (all Thermo Fisher Scientific). Cells were diluted 1:3 for passaging. OS morphology of all primary cell cultures was confirmed by histological analysis performed by an experienced pathologist. For this, the same passage of OS cells used for proliferation assays was paraffin-embedded and characterised by morphological and immunocytochemical analysis. The cells were examined for (1) cell morphology, signs of cell and nuclear atypia, presence of atypical mitoses and osteoid formation (routine HE and Goldner stained sections); (2) lineage differentiation of the tumor cells (standardized immunocytochemical examination); (3) exclusion of contaminating cell populations (standardized immunocytochemical examination) as detailed in Appendix A. Primary cultures were used until passage five. The utilization of the primary cells for molecular analysis and drug screening was approved by the Ethical Commission of Rhineland Palatinate (nr 2019-14373). The commercial available cell line HOS was provided by ATCC. 

### 4.2. Patient History Before Ceritinib/Dasatinib Therapy

A 16-year-old female patient suffered from OS in the left humerus with multiple bipulmonary metastases at diagnosis. First-line chemotherapy according to EURAMOS-1/COSS with surgical treatment of the tumor in the left upper arm and pulmonary metastasis resection was conducted. Almost seven months after therapy had been completed, the first metastatic bipulmonary relapse was diagnosed, which was treated surgically, followed by salvage chemotherapy consisting of carboplatine, etoposide and ifosfamide (over 5 months) [1]. Second relapse was detected immediately after salvage chemotherapy had been started and was treated surgically again. Subsequently, the patient received oral maintenance chemotherapy during which she was tumor-free for more than ten months before further metastases of both lungs and the left thigh (3rd recurrence) were diagnosed. Tumor material from the 3rd recurrence obtained during surgical resection was used for the isolation of primary tumor cells (sample no 386). With the exception of a poor (to reduced) nutritional status, the physical examination showed inconspicuous clinical results with a Karnofsky index of 90% and an ECOG index of 0. This study has been performed in accordance with the ethical standards laid down in the 1964 Declaration of Helsinki and its later amendments. The parents gave their informed consent prior to their inclusion in the study. Formal approval of the local ethics committee for this study was not required, as this was a single case investigation.

### 4.3. DNA Sequencing

PCR products were sequenced by using ABI Prism 3100 Genetic Analyser and the BigDye v3 Terminator Kit (Thermo Fisher, Dreieich, Germany). The sequences were compared to the reference sequence using the Sequencher program (Gene Codes, Ann Arbor, MI, USA). The primers used to sequence *ALK* can be found in the Appendix A.

### 4.4. In Vitro Drug Screening

All inhibitors were commercially purchased (Selleck Chemicals, Houston, TX, USA) and dissolved in dimethyl sulfoxide DMSO (Sigma-Aldrich Co., St. Louis, MO, USA). Cells were plated in triplicates at a density of 3,000 or 5,000 cells/well in a 96-well plate and incubated with the inhibitors for nine days; each plate contained a non-treatment condition, vehicle condition and blank. The medium was completely aspirated every three days and replaced with fresh medium containing fresh solution of inhibitors. Viable cells were quantified using the WST-1 reagent (Roche, Mannheim, Germany).

### 4.5. Histological Analysis

Supposed tumor tissue was obtained from four different pleural sites, immediately frozen using liquid nitrogen and stored at −80 °C. In order to save tissue for further analyses, only one frozen section procedure was performed from each sample to assess the vital tumor content. Hematoxylin and eosin-stained sections were digitalized using a NanoZoomer slide scanner (Hamamatsu Photonics, Hamamatsu, Japan), decolorized and stained with a ready-to-use monoclonal anti-CD68 antibody (IR613, Agilent Technologies, Santa Clara, CA, USA) using an automated staining system in accordance with the manufacturer’s instructions. Tumor regression grading was evaluated according to Huvos et al. and Salzer-Kuntschick et al. [28,29]. The total tissue area and vital tumor area were determined on digitalized hematoxylin and eosin-stained sections using the annotation tool of QuPath [43]. Cell nuclei were automatically detected using a watershed segmentation algorithm based on the hematoxylin optical density. CD68-positive cells were subsequently defined as cells with a 3,3–diaminobenzidine (DAB) mean optical density of more than 0.5.

### 4.6. Phosphorylation Assay

The 386 cells were grown in charcoal-stripped DMEM (high glucose, no glutamine, no phenol red, Sigma-Aldrich Co.) supplemented with 10% human serum, 2 mM L-Glutamin, 1 mM sodium pyruvate and penicillin-streptomycin for 1 day. Charcoal-stripped DMEM was used to eliminate possible residues of factors affecting the IGF pathway. Indeed, IGF2 and IGFBP 6 are commonly depleted by Charcoal-Stripping [44]. The cells were serum-starved for 24 h and then treated with 20 ng/mL IGF2 for 15 min. Cell lysates were loaded on SDS-PAGE gels, followed by blotting to polyvinylidene difluoride membrane (BioRad Laboratories, Inc., Hercules, CA, USA). The following antibodies were obtained from Cell Signaling Technology (Danvers, MA, USA): GAPDH (14C10) (cat # 2118, 1:1000 dilution), IGF1 receptor β (D23H3) XP^®^ (cat # 9750, 1:1000 dilution), Insulin Receptor β (4B8) (cat # 3025, 1:1000 dilution), Phospho-IGF-1 receptor β (Tyr1131)/Insulin Receptor β (Tyr1146) (cat # 3021, 1:1000 dilution), Akt (cat # 9272, 1:1000 dilution), Phospho-Akt (Ser473) (cat # 9271, 1:1000 dilution), Src (32G6) (cat # 2123, 1:1000 dilution) and Phospho-Src Family (Tyr416) (cat # 2698, 1:1000 dilution). The antibodies ERK1/ERK2 (cat # MAB1576, 1:2000 dilution) and Phospho-ERK1/ERK2 (cat # MAB1018, 1:2000 dilution) were purchased from R&D Systems (Minneapolis, MN, USA). Detection was undertaken by SuperSignal™ West Dura Extended Duration Substrate (Thermo Fisher Scientific) and the imaging was performed on Fusion Pulse TS (Vilber Lourmat, Eberhardzell, Germany). Quantification was done with Evolution-Capt Software (Vilber Lourmat) after background substraction with rolling ball. The bands were normalized to control.

### 4.7. Ceritinib and Dasatinib Quantification

Concentrations of ceritinib and dasatinib in plasma and tumor tissue were analyzed by an (ISO) 9001:2015 certified laboratory at the Heidelberg University Hospital. Ceritinib concentrations in tumor tissue and plasma were quantified using UPLC-MS/MS and liquid/liquid extraction techniques as already described in [9]. Dasatinib tumor tissue and plasma concentrations were quantified also using liquid/liquid extraction- and UPLC-MS/MS techniques. The tissue and plasma samples were spiked with the internal standard d8-dasatinib. Subsequently tert-butylmethyl ether was added, shaken for 10 min, and centrifuged (10 min, 3,000 × *g*). The supernatants were evaporated to dryness in a stream of nitrogen at 40 °C, reconstituted by adding LC eluent (150 µL), and injected (10 µL) into the UPLC-MS/MS system, which consisted of an Acquity sample manager, an Acquity solvent manager, and a TQD triple stage quadrupole mass spectrometer (Waters GmbH, Eschborn, Germany). For chromatographic separation, an Acquity BEH Phenyl (1.7 µm; 2.1 × 50 mm) column (Waters GmbH, Eschborn, Germany) at 40 °C was used. The eluent consisted of water, including 5 mM ammonium formate and 5% acetonitrile (A) and acetonitrile (B). For separation a gradient program at 0.5 mL/min was applied. From 0 to 0.5 min, 100% A/0% B was used. From 0.5 to 3.5 min, the ratio was linearly changed to 5% A/95% and the system was equilibrated for 0.5 min with 100% A/0% B. The eluent was introduced directly into the electrospray ion source of the tandem mass spectrometer (MS/MS). The MS/MS transitions monitored in the positive ion mode were m/z 488.32 → m/z 401.18 at 40 V for dasatinib and m/z 496.46 → m/z 405.94 at 42 V for d8-dasatinib. The assay was validated according to common FDA and EMA validation guidelines on bioanalytical method validation. The lower limit of quantification of dasatinib was 1.0 ng/mL in plasma (1.0 ng/g tissue). The calibrated range was 1.0–500 ng/mL plasma or ng/g tissue (2.0–1025 nM) with correlation coefficients >0.995. The overall accuracy varied between –5.0% and +0.3%, and the overall precision ranged from 5.2% to 9.4%.

## 5. Conclusions

In conclusion, our data indicate ceritinib/dasatinib as a new TKI combination to be tested in a clinical study not only for OS patients but also for other IGF and ALK driven tumor entities. By repositioning and prioritizing agents already released, our data may accelerate the access to new therapeutic options. Further studies will be necessary to elucidate the exact mechanism of action of the two drugs in OS cells.

## Figures and Tables

**Figure 1 cancers-12-00793-f001:**
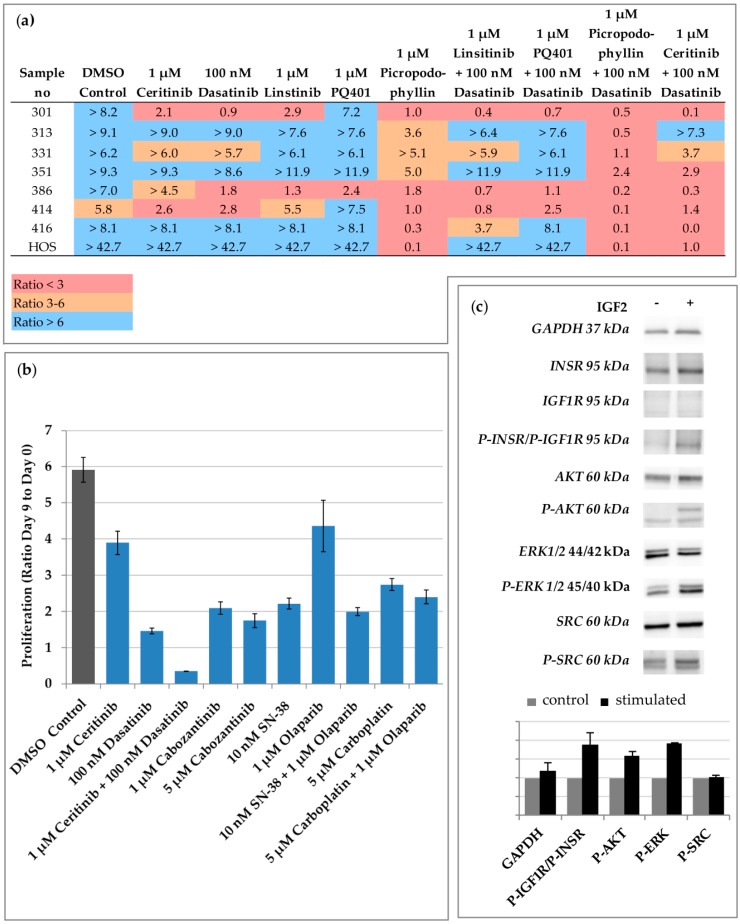
Drug screening identifies ceritinib and dasatinib as potent inhibitors of OS proliferation in vitro. (**a**) Primary tumor cells of six patients (corresponding to seven samples) and the HOS cells were incubated with the indicated inhibitors. After nine days, the cell proliferation was measured. The ratio between the absorbance at day nine and the absorbance at day zero is indicated. At least two independent experiments were used to calculate the ratio and each experiment was done in triplicates. Low proliferation is outlined in red, middle in orange, and strong in blue. (**b**) The primary tumor cells of sample 386 were incubated with the indicated inhibitors. After nine days, the cell proliferation was measured and the ratio between the absorbance at day nine and the absorbance at day zero was calculated. The proliferation experiments were carried out in biological duplicates. Data are represented as the mean ± Standard deviation (SD). (**c**) 386 cells were stimulated after starvation with IGF2. The expression of IGF1R, INSR, AKT serine/threonine kinase 1 (AKT), extracellular-signal regulated kinases (ERK), kinase Src (Src) and their phosphorylated (P) form was analyzed by Western blot with specific antibodies. Glyceraldehyde-3-phosphate dehydrogenase (GAPDH) was used as loading control. A representative experiment of three independent experiments is shown. Densitometric analysis of two independent experiments is shown. The control (unstimulated) condition was normalized to one.

**Figure 2 cancers-12-00793-f002:**
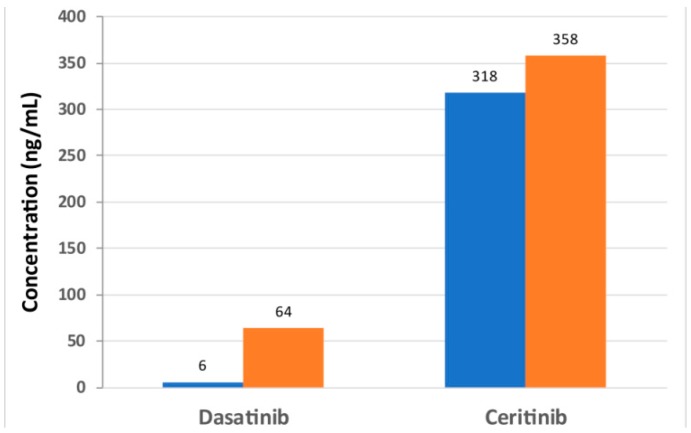
Concentration of ceritinib and dasatinib in plasma. The concentration of ceritinib and dasatinib (in ng/mL) was measured before the application (blue) and four hours after the application (orange).

**Figure 3 cancers-12-00793-f003:**
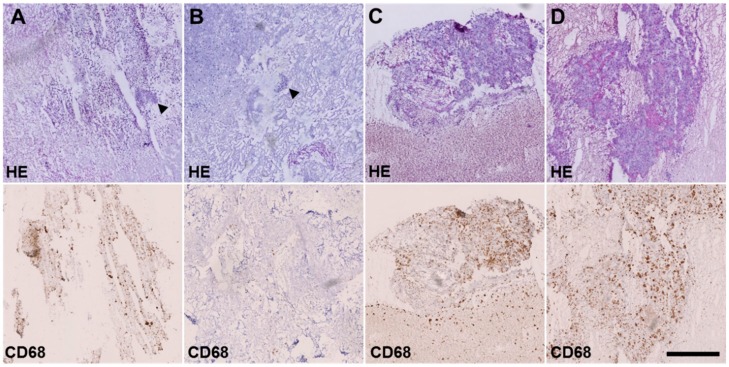
Necrosis and macrophage infiltration after treatment with certinib/dasatinib. Frozen sections from four different pleural sites stained with hematoxylin and eosin (HE, upper row) and CD68 immunohistochemistry (lower row). Conventional histology unveiled extensive necrosis, inflammatory cell infiltration and scattered small tumor cell groups (arrowheads) in samples 1 (**A**) and 2 (**B**). In contrast, vital tumor cell content was increased in sample 3 (**C**) and particularly in sample 4 (**D**). Scale bar: 500 µm.

**Figure 4 cancers-12-00793-f004:**
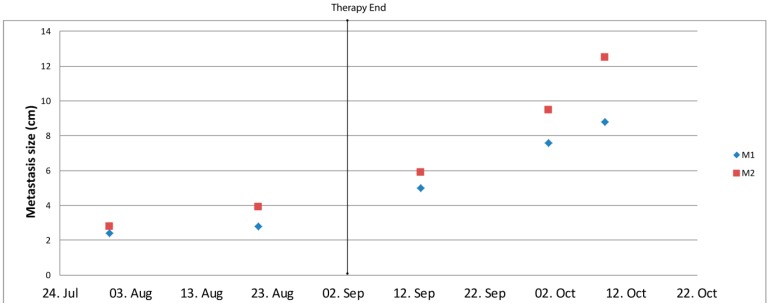
Liver metastases during and after the ceritinib/dasatinib therapy. The size (in cm) of two metastases (M) was monitored by ultrasound examination. The ceritinib/dasatinib combination was assumed until the date indicated by therapy end.

**Table 1 cancers-12-00793-t001:** Osteosarcoma (OS) primary cells used in the study.

Sample no.	Gender	Age	Primary Tumor or Metastasis	Localization
301	m	13	Primary tumor	Distal femur left
313 ^1^	m	18	Metastasis	Lung
331 ^1^	m	18	Metastasis	Lung
351	f	19	Metastasis	Lung
386 ^2^	f	16	Metastasis	Lung
403 ^2^	f	17	Metastasis	Lung
414	m	15	Primary tumor	Proximal tibia left
416	m	15	Primary tumor	Proximal tibia left

^1^ same patient. cells were isolated during surgeries at two different time points; ^2^ same patient; 386 before ceritinib/dasatinib; 403 after ceritinib/dasatinib.

**Table 2 cancers-12-00793-t002:** Correlation of tumor regression and drug penetration in four different pleural specimens.

Analysis	Sample			
	1	2	3	4
Vital tumor area/total area [%]	0.02	0.4	3.6	20.8
Huvos grading *	III	III	II	II
Salzer-Kuntschik grading **	II	II	III	IV
Dasatinib concentration [ng/g]	52.9	12.5	47	0
Ceritinib concentration [ng/g]	1703	1021	743	687
% CD68 cells	26.27%	10.15%	55.46%	31.45%

(*) Huvos et al. 1977 [28], IV: No tumor cells, III, Scattered foci of tumor cells, II: Areas of necrosis and tumor, I: Little or no tumor regression (**) Salzer-Kuntschik et al. 1983 [29], I: No vital tumor cells, II: Single tumor cells or clusters < 0.5 cm, III: Vital tumor cells < 10%, IV: Vital tumor cells 10–50%, Vital tumor cells > 50% and VI: No tumor regression. The % of CD68 positive cells is also indicated.

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
