# Peer review of "Safety and Activity of the Combination of Ceritinib and Dasatinib in Osteosarcoma"

_cancers, 2020, doi:10.3390/cancers12040793_

Round 1
Reviewer 1 Report
The authors present an interesting work in which they describe the effects of treatment with ceritinib and desatinib. However, the work in this form seems an overlap between a clinical case report and a basic research paper.
Major comments:
- How do the authors be sure that cells isolated from lung metastases are OS cells? Please, provide the histological analyses performed. Apart from the pathologist's observations, some specific genes have been analyzed (i.e. ezrin, GJA1, COL1A2 and COL5A2), especially for 386 and 403 cells?
- Which is the meaning of samples 313 and 331 since they gave some opposite results in term of proliferation (fig 1) in some conditions?
- Could the authors report the proliferation data as % of inhibition instead as RATIO?
- Could the authors provide bar graphs for the fig 1c and fig 3 in order to quantify data obtained by WB and by IHC?
- Could the authors provide WB experiments after treating OS cells with ceritinib + desatinib (pSrc, pFAK, p-P130CAS, pJun)?
6. Could the authors performed migration/adhesion in vitro experiments with OS cells, in the presence of ceritinib + desatinib?
Minor comments:
Label for y axis in fig 2 is missing
Reviewer 2 Report
This study aimed to evaluate new therapeutic option based on IGF/IGFR pathway targeting in osteosarcoma. The manuscript describes the effects of the combination of dasatinib to several IGF1R inhibitors (linsitinib, picropodophyllin, PQ401) or to ceritinib on primary tumor cells derived from patients and on an established cell line (HOS). A second part of the manuscript described a therapy protocol combining dasatinib and ceritinib applied to an individual patient suffering from osteosarcoma with multiple lung metastases and a long history of surgical resection/chemotherapy cycles.
Specific comments:
What are the relative expression levels/activation states of IGF1R or downstream targets in the panel of primary tumor cells? This should reinforce the rational for testing IGF1R inhibitors.
How many medium changes were performed between seeding and WST-1 assay at day 9? How stable was the drug concentration along the experiment? Were dead cells discarded regularly?
The sentence line 96 “Clinical relevant concentrations were applied…” is not informative. The determination of the concentrations for each drug must be further detailed. Is it related to IC50 values? PPP drug at 1 µM seems to be highly active even on HOS cells. Lower concentration should be tested to confirm a possible additional effect with other compounds.
What is the rational for a charcoal-stripping of the DMEM and not of the serum? Is this restricted to the phosphorylation assay or for the culture maintenance and all experiments?
The evaluation of the IGF2-dependent induction of the IGF1R signaling pathway could be completed by testing the ceritinib / dasatinib combination.
The dilution ratio for the passages of primary cultures (line 306) must be indicated.
Globally the Y-axis legends are poorly descriptive and should be revised.
Round 2
Reviewer 1 Report
The authors have reply to all comments and requests by implementing the paper with new results, where possible, also given the few days provided by the editor to respond. To my concern the paper can be accepted.